# The BiteBarrier transfluthrin emanator demonstrates significant protection against susceptible and resistant malaria and arbovirus vectors in semi-field trials in Tanzania

**Masudi Suleiman Maasayi[1,2], Johnson Kyeba Swai[1,3,4], Joseph Barnabas Muganga[1], Jason Moore[1,3], Jennifer Claire Stevenson[1,3,4], Michael Coleman[5]\*, Neil Francis Lobo[6], Sarah Jane Moore[1,2,3,4], Mgeni Mohamed Tambwe[1,3,4]**

1 Vector Control Product Testing Unit, Environmental Health and Ecological Science Department, Ifakara Health Institute, Bagamoyo, Tanzania, 2 School of Life Sciences and Bioengineering, The Nelson Mandela African Institution of Science and Technology (NM-AIST), Arusha, Tanzania, 3 Vector Biology Unit, Epidemiology and Public Health Department, Swiss Tropical and Public Health Institute, Kreuzstrasse 2, Allschwil, Basel, Switzerland, 4 University of Basel, Petersplatz, Basel, Switzerland, 5 Liverpool School of Tropical Medicine, Liverpool, United Kingdom, 6 Eck Institute for Global Health, University of Notre Dame, Notre Dame, Indiana, United States of America

\* michael.coleman@lstmed.ac.uk

## Abstract

Controlling mosquito-borne diseases is becoming increasingly challenging due to factors such as insecticide resistance and shifts in mosquito behavior. The increasing proportion of early evening, morning, and outdoor biting reduces the effectiveness of core interventions like bed nets, which mainly protect people while sleeping indoors. In response, spatial emanators that release volatile active ingredients into the surrounding air to reduce human-vector contact offer a scalable, complementary strategy. This study evaluated the impact of BiteBarrier, a transfluthrin-based spatial emanator, over eight weeks of aging against multiple mosquito species in a semi-field system simulating both indoor and outdoor settings. We assessed the protective efficacy using both landing rate and feeding success methods across five mosquito species, including pyrethroid susceptible *Anopheles gambiae* sensu stricto (s.s.) and *Aedes aegypti*; *An. gambiae* s.s. with knock down resistance (KDR); and pyrethroid resistant *An. funestus* and *Culex quinquefasciatus* with upregulation of mixed function oxidases. The results show that the feeding endpoint provides more robust estimates of protective efficacy compared to the landing endpoint. The BiteBarrier provided over 93% (95% CI: 92–93) protection indoors and 80% (95% CI: 78–81) outdoors against mosquito bites and substantial mortality 47% (95% CI: 43–53) indoor and 26% (95% CI: 22–30) outdoors, regardless of mosquito species or resistance status. Overall, the BiteBarrier shows potential as a tool for reducing mosquito bites and vectorial capacity, offering protection over at least eight weeks of use for both indoor and outdoor environments.

**Data availability statement:** Data used to generate conclusions of this study are available from the supplementary information.

**Funding:** This study was sponsored by the Department of the Army, U.S. Army Contracting Command, Aberdeen Proving Ground, Edgewood Contracting Division, Ft Detrick MD [under Deployed Warfighter Protection (DWFP) Program Grant W911SR2210002] to MC. The funders had no role in study design, data collection and analysis, decision to publish, or preparation of the manuscript.

**Competing interests:** The authors have declared that no competing interests exist.

**Abbreviations:** SEs, Spatial emanators; PE, Protective efficacy; BB, BiteBarrier; SFS, Semi-field system.

## Background

The transmission dynamics and geographic risk of vector-borne diseases, such as malaria and arboviruses, are shifting in response to changing settlement patterns, economic activities, and environmental changes [1,2]. Biological and behavioral challenges, including mosquito resistance to insecticides and changes in biting patterns, undermine current control efforts [3]. To accelerate malaria control and address the rising threat of arboviral infections, new vector control tools are urgently needed.

Spatial emanators (SEs) offer promise as scalable solutions to reducing human-mosquito contact and ultimately reducing disease transmission [4,5]. These emanators release active ingredients into the air, inducing various behavioral responses in exposed mosquitoes including excito-repellency, interruption of host-seeking and feeding, incapacitation and mortality [6]. Numerous semi-field [7–9] and small-scale field studies [9–12] have demonstrated the efficacy of SEs in reducing mosquito landing, blood-feeding, and survival. Additionally, they have been observed to have public health benefits by reducing transmission of malaria [5,13–16] and *Aedes*-borne viruses [17]. Further ongoing randomized controlled trials (RCTs) in various ecological settings are evaluating the potential of SEs to reduce malaria [18] and arboviruses [19] transmission and the public health benefits of their operational implementation.

It is a World Health Organization (WHO) requirement that new vector control products undergo rigorous testing ranging from laboratory to semi-field and small to large-scale field studies to demonstrate safety, and efficacy before they are recommended for public use. Semi-field studies bridge the gap between laboratory findings and field efficacy studies, evaluating entomological efficacy against free-flying laboratory reared mosquitoes under controlled simulated indoor or outdoor conditions [20]. Since laboratory reared mosquitoes are disease-free, the semi-field system (SFS) allows investigators to safely test interventions even against dengue vectors. This approach allows mosquitoes to blood feed, which may not be feasible in certain field studies. It also allows to use mosquitoes with a known age, physiological and resistance status enabling comprehension of impact over a wide array of selected biological traits that may be present in field transmission systems. Furthermore, the SFS allows for release and recapture of mosquitoes after they have interacted with humans in the presence of an intervention so that additional modes of action, including post-exposure mortality may be evaluated.

This study assessed the protective efficacy (PE) of the BiteBarrier (BB) transfluthrin emanator against East African malaria and arbovirus vectors in semi-field simulations of indoor and outdoor contexts. Specifically, the study aimed to evaluate the BB over an eight-week period by: 1) comparing PE measured if mosquitoes are captured when landing (landing method) or if free flying mosquitoes are allowed to blood feed (feeding method), 2) measuring the indoor and outdoor PE of the BB and, 3) estimate the effect of BB on mosquito mortality.

## Methods

### Study setting

The study was conducted in the SFS at the Ifakara Health Institute (IHI) in Bagamoyo, Tanzania. The SFS is described in detail elsewhere [21] and was modified to accommodate the objectives of the experiment (Fig 1. A). These modifications involved further dividing the larger SFS compartments using Plywood and heavy-duty polyurethane sheets to make four independent chambers each measuring 10 x 9 m. This allowed for both indoor and outdoor experiments to be conducted simultaneously, with each treatment and its control allocated to an independent chamber. During the experiments, the median environmental conditions were 24.4°C (24–30°C) and 85.9% (62–100%) relative humidity. Wind speed was measured in the morning before the experiments and was 0.0 m/s.

### Test systems (mosquitoes)

Laboratory-reared 3–5 days old pyrethroid-susceptible *Anopheles gambiae* s.s., *An. gambiae* s.s. with knock down resistance (KDR)*,* pyrethroid-resistant *An. funestus,* pyrethroid-resistant *Culex quinquefasciatus* and pyrethroid-susceptible *Aedes aegypti* female mosquitoes were used in the experiments. Detailed susceptibility profiles for each are shown in S2 Table. The mosquitoes were sugar starved for six hours and acclimatized in releasing cages in the control chambers for 30 minutes before the tests. The colony was maintained by feeding larvae on Tetramin® fish food and adults on 10% glucose solution *ad libitum* and blood for egg laying. Temperature and relative humidity within the insectary were maintained following MR4 guidelines at 27±2ºC and 70±20%, respectively [22]. As all mosquito strains were released together, *An. gambiae* s.s. (KDR) were marked with fluorescent dye to distinguish them from *An. gambiae* s.s. The coloring procedure has been optimized and has been shown to have no significant effect on mosquito fitness or survival [23].

### The BiteBarrier

The BiteBarrier (BB) is a novel passive emanator dosed with 1.5 mg of transfluthrin on a non-woven substrate material that consists of two 24 x 28 cm sheets (a total area of 1,344 cm$^2$). The BB emanators were aged over eight weeks by hanging them under temperature-controlled conditions similar to those in tropical regions (24.5–27.5°C). After aging, the BBs were wrapped in aluminum foil, placed in a sealed plastic bag, and stored in a cool dry room with a temperature not exceeding 20ºC. The temperature and humidity were monitored daily during the aging and storage process using Tinytag climatic logger (Gemini Data Loggers Ltd, Chichester, UK).

### Study design

A series of two partially balanced 2 x 2 Latin square design experiments were conducted in four SFS chambers. The chambers are divided using high density polypropylene sheeting to ensure independence of observations and prevent contamination between chambers. Two chambers were designated for simulated indoor and two for simulated outdoor settings. For each setting, one chamber served as the treatment (with BB installed) and one as a negative control (no BB). The control chambers were located adjacent to the treatment chambers, with wind directional movement from the control to the treatment chambers. To prevent cross-contamination, treatment and control chambers remained fixed throughout the study. Four male volunteers (two for outdoor and two for indoor), aged 25–40, non-smokers, and non-drinkers, participated in the study after providing written informed consent. During the one-hour exposure period, mosquitoes were allowed to interact with the volunteers, who had only the area between the knee and ankle uncovered (Fig 1. B, D). This standardizes the areas for mosquito landing and feeding. Volunteers rotated between treatment and control conditions within their assigned context (indoors or outdoors) to minimize bias. Before the study, we conducted preliminary assessments of mosquito attractiveness and found no significant differences among the volunteers. The experiment was replicated 30 times over five rounds and 30 experimental days. Each round lasted six days, followed by a three-day washout

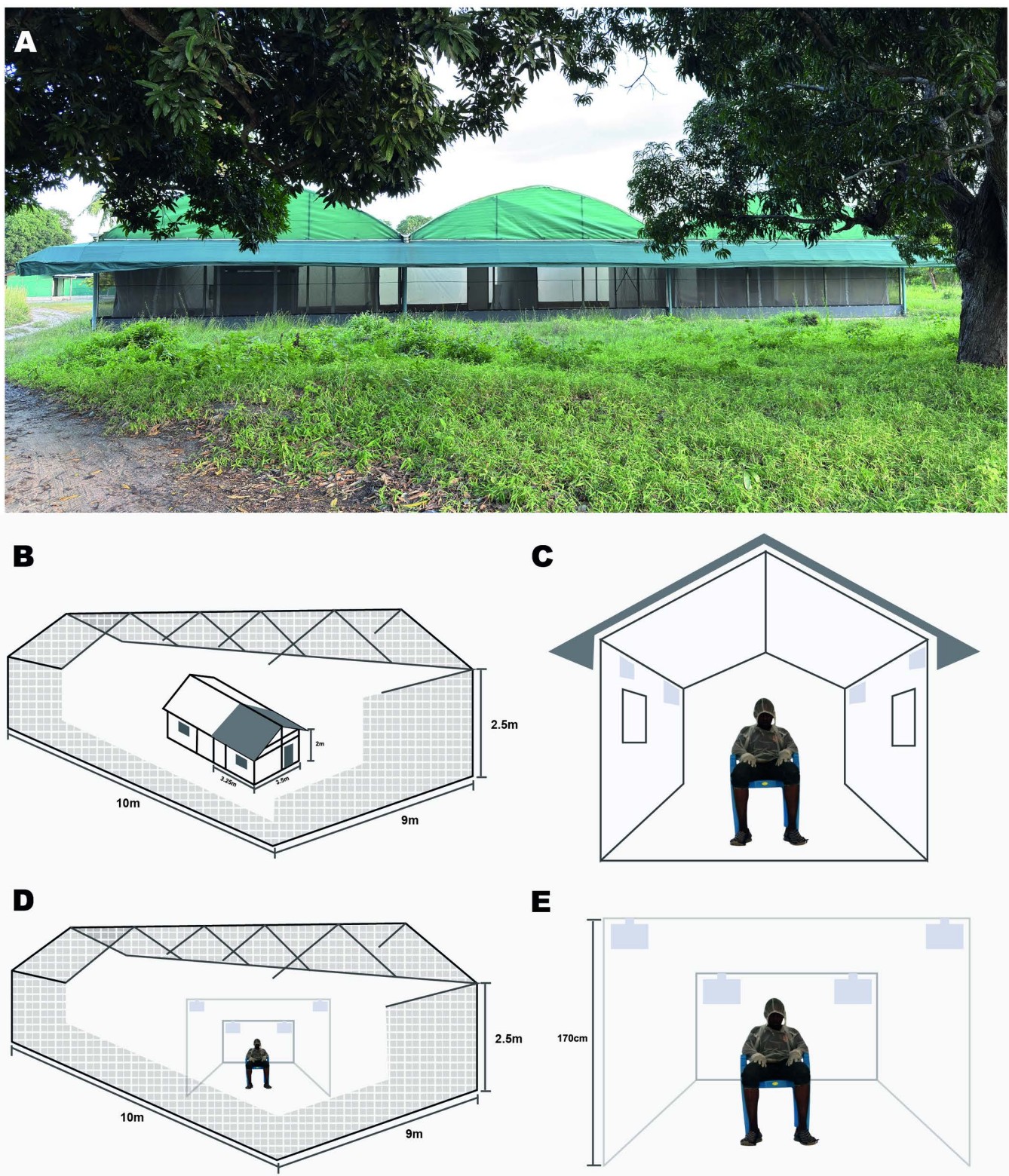

**Fig 1. Schematic representation of the semi-field experiments.** The outside view of the semi-field system **(A)**, Indoor evaluation **(B, C)** and outdoor evaluation **(D, E)**.

period. On the third washout day, mosquitoes were released into all chambers without any treatment to assess residual activity, and landing rates were similar across all chambers.

## Indoor evaluation of the BiteBarrier

To simulate an indoor setting, two Ifakara experimental huts (one hut per chamber) were installed in the two SFS chambers. The huts are divided with a fully sealed plywood wall to make two huts. For this experiment, only one side of each hut was used (Fig 1. A, B), while the other side was sealed. The dimensions of the huts are 3.25 x 3.5 x 2 m (length x width x height) with a gabled roof of 0.5 m apex and volume of 25.59 m³. Each hut has 10 cm-wide eave gaps on three sides fitted with baffles that allow mosquitoes to enter freely. In one experimental hut, one BB sheet was hung in each corner (170 cm), while the other hut served as a negative control with no BBs. Each day, the BB devices were set up two hours before mosquito release to allow the transfluthrin to diffuse. Volunteers sat at the center of the hut while mosquitoes were released outside the hut from two points (Fig 1. B,C). Experiments with *Anopheles* and *Culex* mosquitoes were conducted from 18:30–22:30. Each session included one-hour landing replicate with 80 mosquitoes of each strain released (18:30–19:30), followed by one hour for collection (removing all mosquitoes in the hut and chamber). Another one-hour feeding replicate was conducted with another batch of 80 mosquitoes of each strain released (20:30–21:30) followed by another hour of collection. For *Ae. aegypti*, experiments were conducted from 6:30–10:30, following similar pattern. Landing and feeding experiment times were alternated after every three replicates to control for temporal bias. After each replicate, BB devices were wrapped in aluminum foil and stored at 20°C to prevent further evaporation. Eighty per strain for each replicate to ensure that at least 40 mosquitoes enter the huts to maintain the study power. This decision was based on preliminary experiments which showed that at least 50% of the mosquitoes could enter indoors (inside the hut) after one hour following release in absence of any treatment. Environmental conditions in the SFS were monitored using a Tinytag climatic logger (Gemini Data Loggers Ltd, Chichester, UK).

## Outdoor evaluation of the BiteBarrier

Outdoor experiments were conducted in a large net cage measuring 10 x 9 m. In the treatment chamber, one volunteer sat two meters away from four BB sheets hung at a height of 170 cm (Fig 1. D,E). In the control chamber, a volunteer was similarly positioned, but no BB devices were installed. Forty mosquitoes of each strain were released for each landing and feeding replicate. Other experimental procedures were maintained as for the indoor experiment.

## Procedures for landing experiment

Volunteers used mouth aspirators to collect mosquitoes landing between the knee and ankle (Fig 1). Collections were conducted in 15 minutes intervals, with mosquitoes placed in separate paper cups for each period. After collection, cups were sealed in plastic containers to prevent additional transfluthrin exposure. The remaining mosquitoes were collected after one hour using Prokopack aspirators ((John W Hock, Gainesville, FL), and all samples were transported to the insectary for sorting and recording. Mosquitoes collected through landing were held for 72 hours with access to 10% sucrose solution to observe delayed mortality.

## Procedures for feeding experiment

Volunteers sat on chairs, allowing mosquitoes to feed on their exposed legs (Fig 1). After the exposure period, mosquitoes were collected for one hour from the floor, walls, roof in the net cages and/or huts using mouth aspirators. Mosquitoes were placed in paper cups (maximum 25 per cup) to reduce density-related mortality. All collected mosquitoes were taken to environmentally controlled insectaries for sorting and held for 72 hours with access to 10% sucrose solution to observe delayed mortality.

## Ethics declarations

The study was approved by the Ifakara Health Institute Review Board with certificate number: IHI/IRB/EXT/No: 01–2024, the National Institute for Medical Research-Tanzania (NIMR) with a certificate number: NIMR/HQ/R.8b/Vo1.I/1165, with ethics and sponsorship from Liverpool School of Tropical Medicine (LSTM) protocol number 22–067. Study participants were IHI entomology staff who were fully informed of the risks and voluntarily provided informed consent. Their employment was not contingent on participation in the study.

## Statistical analysis

Statistical analysis was conducted using Stata 17. Descriptive statistics were used to explore the data. To compare between the landing and feeding methods in the treatment and control, the number of mosquitoes collected in the landing experiment and the number fed mosquitoes in the feeding experiment were merged forming a single variable named "recaptured". Recaptured mosquitoes were modelled independently for each mosquito species and overall (including all species) using mixed effects logistic regression with binomial distributions (recaptured/recovered) with a logit function. The method of collection (landing vs feeding), treatment (control vs treatment), context (indoor vs outdoor), volunteer, chamber and day, were treated as independent categorical fixed effects. Humidity and temperature were added to the model as continuous variables.

To estimate the protective efficacy of the BB using feeding method, mixed effects logistic regression following binomial distributions (fed/recovered) with a logit function was used. Each species and context were analyzed independently. Treatment (control vs treatment), volunteer, chamber and day were treated as independent categorical fixed effects. Similarly, temperature and humidity were added to the model as continuous variables. The PE (reduction in the Odds of feeding) of the BB was calculated from the odds ratios (OR) obtained from the model using the formula $(1-OR)*100$.

Additionally, the effect of the BB on delayed 72 hours mortality was estimated using data obtained from feeding experiment using mixed effects logistic regression with binomial distributions (dead/recovered). Each species and context were analyzed independently. Treatment (control vs treatment), volunteer, chamber and day were treated as independent categorical fixed effects, whereas temperature and humidity were added to the model as continuous variables. Due to the low number of dead mosquitoes in the control arm and to ensure model convergence, 1 was added to all observations of dead mosquitoes in the control and the treatment arms.

## Results

### Mosquito recovery in the semi-field system

Overall recovery rates, defined as the proportion of released mosquitoes that were recaptured were consistent across landing and feeding methods. Indoors (the experimental hut experiment), recovery rates in the treatment chambers were 76% and 77%, compared to 92% and 93% in the controls for the landing and feeding methods, respectively. Outdoors, recovery rates in the treatment chambers were 99% and 100% similar to those in the control chambers 100% and 100% for landing and feeding methods, respectively. The recovery rates were consistent for each species and methods (S1 Table).

### Landing rate and blood feeding success

The landing rate, which is defined as the proportion of recaptured mosquitoes caught in the landing experiment, exhibited variation depending on the specific context in which the data were collected. Indoors, the landing rate was 14.63% (95% CI: 12.07–17.18) in the treatment and 62.31% (95% CI: 57.5–67.09) in the control chamber. Outdoors, the landing rate was 43.36% (95% CI: 39.63–47.08) in the treatment and 65.93% (95% CI: 61.97–69.91) in the control chamber. Similarly, feeding success, defined as the proportion of recaptured mosquitoes that were blood-fed also varied by context. Indoors,

feeding success was 7.92% (95% CI: 5.97–9.87) in the treatment and 53.86% (95% CI: 49.96–57.75) in the control. Outdoors, feeding success was 43.57% (95% CI: 39.29–47.84) in the treatment and 77.39% (95% CI: 73.46–81.32) in the control chambers.

### Comparison of protective efficacy estimates between landing and feeding methods

Across all mosquito species, the odds of feeding was lower compared to the landing [OR = 0.83, 95% CI: (0.80–0.86), P<0.002]. However, this varied for each species (Table 1). For *Anopheles gambiae* s.s. there was no difference measured by either method. Feeding success was significantly lower than landing rate for *An. gambiae* s.s. (KDR). Conversely, feeding success was significantly higher than landing rate for *An. funestus*, *Cx. quinquefasciatus* and *Ae. aegypti.* Therefore, in subsequent analysis we opted to use data from the feeding experiment only as this endpoint captures both mosquito attack and feeding behavior, both of which are modified by volatile pyrethroids.

### Protective efficacy of the BiteBarrier transfluthrin emanator measured by blood-feeding method

Overall, the BB gave similar high protection against all mosquito strains regardless of species or resistance status. This included all three Afrotropical malaria vectors (susceptible *An. gambiae* s.s., *An. gambiae* s.s. KDR, as well as resistant *An. funestus* that has upregulation of mixed function oxidases) and arbovirus vectors (resistant *Cx. quinquefasciatus* that has upregulation of mixed function oxidases and susceptible *Ae. aegypti*) (Table 2). Protective efficacy was higher indoors 93% (95% CI: 0.92–0.93) than outdoors 80% (95% CI: 0.78–0.81) (Table 2).

### Effect of the BiteBarrier on mosquito mortality

Overall, mortality in the control arm throughout the experiment was below 5% for each mosquito species. The BB induced substantial mortality with higher mortality observed indoors than outdoors. Mortality was higher among malaria vectors than arbovirus vectors (Table 3) and appeared to be related to susceptibility levels (S2 Table). The highest mortality rate was observed in the susceptible *Anopheles* strain while the lowest mortality was observed in highly resistant *Cx. quinquefasciatus* (Table 3).

**Table 1. Comparison between landing and feeding methods for measuring the protective efficacy of the BiteBarrier transfluthrin emanator.**

| Mosquito species | Method | n/N | OR (95% CI) | P value |
|---|---|---|---|---|
| *An. gambiae* s.s. | landing | 475/2,673 | 1 | |
| | Feeding | 447/2,617 | 1.01 (0.92–1.10) | 0.089 |
| *An. gambiae* s.s. KDR | landing | 588/3,179 | 1 | |
| | Feeding | 483/3,155 | 1.11 (1.02–1.20) | 0.011 |
| *An. funestus* | landing | 679/2,615 | 1 | |
| | Feeding | 550/2,755 | 0.78 (0.72–0.85) | <0.002 |
| *Cx. quinquefasciatus* | landing | 1131/3,671 | 1 | |
| | Feeding | 778/3,502 | 0.52 (0.48–0.56) | <0.002 |
| *Ae. aegypti* | landing | 1142/3,230 | 1 | |
| | Feeding | 1036/3,069 | 0.89 (0.83–0.97) | 0.007 |
| Overall | landing | 4,015/15,368 | 1 | |
| | Feeding | 3,294/15,098 | 0.83 (0.80–0.86) | <0.002 |

N refers total number of mosquitoes of each species that were recovered in the treatment chambers while n refers total number of mosquitoes that landed or fed in the treatment chambers; Odds ratios (OR) and P-values were obtained from mixed effect logistic regression model adjusted for the effect of method, treatment, context, volunteer, chamber, experimental night, temperature and humidity, the outcome was "recaptured" defined as total number of mosquitoes of each species successfully landed or fed on study volunteers.

**Table 2. Protective efficacy of the BiteBarrier transfluthrin emanator against *Anopheles*, *Culex* and *Aedes* mosquito species measured using feeding method.**

| Mosquito species | Treatment | Indoor | | | Outdoor | | |
|---|---|---|---|---|---|---|---|
| | | n/N | OR (95% CI) | %PE (1-OR)*100 | n/N | OR (95% CI) | %PE (1-OR)*100 |
| *An. gambiae* s.s. | Control | 819/1,773 | 1 | | 711/1,055 | 1 | |
| | BB | 70/1,540 | 0.05 (0.04–0.07) | 95 (93–96) | 377/1,077 | 0.25 (0.21–0.30) | 75 (70–79) |
| *An. gambiae* s.s. KDR | Control | 897/2,348 | 1 | | 1004/1,341 | 1 | |
| | BB | 136/1,891 | 0.11 (0.09–0.14) | 89 (86–91) | 347/1,264 | 0.11 (0.09–0.12) | 89 (88–91) |
| *An. funestus* | Control | 1330/2,212 | 1 | | 1011/1,214 | 1 | |
| | BB | 103/1,697 | 0.04 (0.03–0.05) | 96 (95–97) | 447/1,058 | 0.14 (0.12–0.17) | 86 (83–88) |
| *Cx. quinquefasciatus* | Control | 1329/2,331 | 1 | | 1051/1,456 | 1 | |
| | BB | 204/2,123 | 0.07 (0.06–0.08) | 93 (92–94) | 574/1,379 | 0.27 (0.23 −0.31) | 73 (69–77) |
| *Ae. aegypti* | Control | 1556/2,336 | 1 | | 1143/1,263 | 1 | |
| | BB | 213/1,855 | 0.06 (0.05–0.07) | 94 (93–95) | 823/1,214 | 0.21 (0.17–0.27) | 79 (73–83) |
| Overall | Control | 5931/11,000 | | | 4,920/6,000 | | |
| | BB | 726/9,106 | 0.07 (0.07–0.08) | 93 (92–93) | 2,568/5,992 | 0.20 (0.19–0.22) | 80 (78–81) |

N refers total number of mosquitoes of each species that were recovered in the semi-field system while n refers total number blood-fed mosquitoes; Odds ratios (OR) and p-value were obtained from mixed effect logistic regression model adjusted for the effect of treatment, volunteer, chamber, experimental night, temperature and humidity; All P < 0.002. PE is the reduction in the odds of mosquitoes feeding.

**Table 3. Effect of the BiteBarrier transfluthrin emanator on mortality of exposed mosquitoes.**

| Mosquito species | Treatment | Indoor | | Outdoor | |
|---|---|---|---|---|---|
| | | Mean (95% CI) | OR (95% CI) | Mean (95% CI) | OR (95% CI) |
| *An. gambiae* s.s. | Control | 1.2 (0.43–1.98) | 1 | 4.71 (0.48–9.91) | 1 |
| | BB | 73.37 (61.29–85.4) | 3.07 (1.51–6.24) | 52.16 (42.65–61.68) | 2.18 (1.14–4.15) |
| *An. Gambiae* s.s. KDR | Control | 0.05 (0.02–0.14) | 1 | 1.07 (0.17–1.97) | 1 |
| | BB | 47.33 (34.89–59.77) | 2.26 (1.46–3.49) | 20.65 (15.17–26.14) | 1.70 (1.12–2.59) |
| *An. funestus* | Control | 1.17 (0.38–1.96) | 1 | 2.05 (0.74–3.36) | 1 |
| | BB | 54.09 (44.08–64.10) | 1.96 (1.30–2.94) | 29.53 (24.43–34.64) | 1.57 (1.06–2.35) |
| *Cx. quinquefasciatus* | Control | 1.30 (0.15–2.45) | 1 | 0.93 (0.36–2.22) | 1 |
| | BB | 19.08 (11.22–26.93) | 1.55 (1.02–2.34) | 15.22 (10.24–20.19) | 1.79 (1.17–2.73) |
| *Ae. aegypti* | Control | 1.39 (0.05–2.73) | 1 | 1.63 (0.38–2.89) | 1 |
| | BB | 47.18 (34.01–60.37) | 2.90 (1.46–5.74) | 14.98 (8.22–21.73) | 2.11 (1.07–4.15) |
| Overall | Control | 1.02 (0.61–1.41) | 1 | 2.08 (0.97–3.19) | 1 |
| | BB | 47.84 (43.26–53.41) | 1.80 (1.49–2.16)* | 26.51 (22.91–30.11) | 1.54 (1.28–1.84)* |

Mean refers to as arithmetic mean of proportion of mosquitoes of each species died per treatment arm; Odds ratio (OR) and P-values were obtained from mixed effect logistic regression model adjusted for the effect of treatment, volunteer, chamber, experimental night, temperature and humidity; *P value <0.002, all other P<0.018

## Discussion

This study demonstrated the efficacy of the BiteBarrier (BB), a transfluthrin-based emanator in reducing potential mosquito bites and survival across multiple species in simulated indoor and outdoor contexts. We found that BB reduced the odds of blood-feeding by 93% (95% CI: 92–93) indoors and 80% (95% CI: 78–81) outdoors and induced significant mortality by 47% (95% CI: 43–53) indoor and 26% (95% CI: 22–30) outdoors against all mosquito species tested.

Previous studies on volatile pyrethroids have used human landing catch (HLC) as a proxy for blood-feeding [24,25] or allowed mosquitoes to bite freely to measure feeding inhibition [9]. Some studies found similar efficacy [10,26], while others found different but reasonable agreement between the two methods [8]. In this current study, feeding method provided a significantly higher estimate of PE of the BB across multiple species, particularly *Cx. quinquefasciatus*, *An. funestus* and *Ae. aegypti*. However, for *An. gambiae* s.s., both methods yielded similar results, while landing method estimated greater PE for *An. gambiae* s.s. (KDR). In a previous study [8] using transfluthrin hessian strips that conducted a similar comparison the results agree closely for *An. funestus*, (OR: 0.78 (95% CI: 0.72–0.85 in the current work vs. (RR: 0.75 (95% CI: 0.63–0.89 in the previous study), and *An. gambiae* s.s. (KDR), (OR: 1.11 (95% CI: 1.02–1.20 in this work) vs. (RR: 0.97 (95% CI: 0.80–1.17 in the previous study). However, the difference between methods was less pronounced for *An. gambiae* s.s., (OR: 1.01 (95% CI: 0.92–1.10 in the current study) vs. (RR: 0.77 (95% CI: 0.63–0.94 in the earlier work). Since not all mosquitoes landing after exposure to transfluthrin feed, as indicated by higher PE in feeding experiments, measuring blood-feeding is important for understanding the potential for pathogen acquisition by mosquitoes. However, we acknowledge that, while landing rate is not equivalent to feeding, remains epidemiologically important for assessing human exposure risks. Although, both feeding and landing assays provide complimentary information, given the mode of action of transfluthrin we recommend blood-feeding as a more accurate measure of PE in proof-of-concept studies under controlled settings. In contrast, the HLC while involving some exposure to probing and potential pathogen transmission, remains a practical and widely used proxy in field evaluations especially where ethical, logistical and regulatory constrains limit the use of blood feeding assays [8,10,27].

The recovery rate of mosquitoes in the SFS (S1 Table) was consistent across both methods, indicating that differences in landing and feeding behavior were not due to mosquito density variations. Indoor and outdoor recovery rates between treatment and control chambers were similar. The slightly lower recovery rates in the indoor treatment chambers may be due to transfluthrin-induced disorientation within the enclosed hut environment, making some mosquitoes less likely to be recaptured. Overall, in a few instances, recovery rates were below or exceeded the estimated release mosquito count, which is expected especially when working with multiple species and complex setups. Instances where the number of recovered mosquitoes exceeded the release count were rare and did not significantly affect the overall recovery patterns across replicates. We acknowledge that, despite a one-hour collection period, some mosquitoes may temporarily remain hidden in the large semi-field system despite that we lowered the roof and covered the floor with white tarpaulin to enhance visibility. Similar scenario has been observed in previous semi-field studies [7,28], highlighting the importance of using recovered mosquitoes as the denominator in analysis for studies conducted in larger semi-field chambers.

Notably, our findings demonstrated that the BB significantly reduces feeding success across all mosquito species. Overall, PE ranged from 92–93% indoors and 78–81% outdoors across species, irrespective of resistance status, reinforcing evidence of transfluthrin's effectiveness against both pyrethroid-susceptible and resistant mosquitoes [5,29]. The higher PE observed indoors is likely due to the higher concentration of transfluthrin inside the hut. Outdoor efficacy could potentially be enhanced by increasing the number of emanators. In a previous semi-field study using the BB, Burton *et al*., [7] reported over 40% reduced host-seeking by *An. gambiae* s.s. with fresh devices, with efficacy slightly declining but persisting over five weeks. Similarly, Vajda *et al*., [28] observed over 50% reduction in the odds of *An. minimus* landings in the BB treated arm deployed for 30 days. Field studies further confirmed effectiveness of the BB showing over 94% protection against *Anopheles* landings in Cambodia [11]. These findings highlight the BB's potential for protecting people in diverse contexts and the utility of SEs to bridging protection gaps, such as indoor biting when people are awake and not using insecticide-treated nets (ITNs) or sleeping but not using ITNs and outdoor biting [30,31]. Additionally, SEs are suitable for protecting individuals in humanitarian or other emergency settings and mobile communities such as forest workers who live in temporary structures [32].

Furthermore, transfluthrin exposure led to significant mosquito mortality especially indoors, aligning with previous studies [7,9,29,33,34]. The high mortality combined with reduction in feeding, highlights the intervention's potential to reduce mosquito populations and disrupt transmission cycles [4,35,36]. Mortality was highest among susceptible and resistant *Anopheles* species and susceptible *Ae. aegypti*, while *Cx. quinquefasciatus*, highly resistant to pyrethroids, showed lower mortality. The observed lower mortality for *Cx. quinquefasciatus* may be due to the high level of pyrethroid resistance observed in this strain (S2 Table), and likelihood of other resistance mechanisms. However, resistance to transfluthrin for this strain is not fully established. Therefore, further experiments are warranted using recommended discriminating doses and determine any potential cross-resistance.

The design of this study, involved only one-hour of mosquito exposure, a relatively shorter period compared to other semi-field experiments where exposure lasted several hours at night. Still, it may not fully, replicate real-world conditions, where host-seeking mosquitoes often exit quickly the treated space due to irritation, disorientation, or repellency. Shorter exposure times combined with repeated mosquito release and collection are recommended to better mimic field conditions [7]. Additionally, while the SFS allows for controlled experimental variables and standardized data collected, it does not fully capture the complexities of real-world environments. Variables such as wind and other environmental conditions could significantly influence the performance of SEs [33,37]. Even so, the inclusion of multiple mosquito species with different susceptibility levels offered a comprehensive evaluation, ensuring that the findings reflect diverse ecological scenarios. Furthermore, SFS studies are valuable as part of a staged process for product evaluation, providing data for proof-of-concept studies and optimizing products or deployment options. For example, our study has provided valuable data that can be used to inform onward experimental hut or field studies.

## Conclusions

Overall, this study provides evidence for the efficacy of the BiteBarrier spatial emanator over eight weeks of deployment in semi-field-controlled settings in reducing mosquito bites and survival for a range of species, including resistant populations. The results highlight the potential of spatial emanators as a valuable addition to the current arsenal of vector control tools, even in areas where resistance to pyrethroids is prevalent or other conventional tools may not be feasible. Further longitudinal research beyond the eight weeks of aging is needed to explore its longer-term effectiveness and public health impact in real-world settings.

## Supporting information

**S1 Table. Mosquito recovery rate in the semi-field system for landing and feeding experiments.**
(DOCX)

**S2 Table. Resistance profile for the different mosquito species tested between December 2023 and February 2024.**
(DOCX)

**S1 Data. Dataset.**
(XLS)

**S1 File. Inclusivity-in-global-research-questionnaire.**
(DOCX)

## Acknowledgments

We acknowledge the insectary, testing and administration teams at the Vector Control Product Testing Unit (VCPTU) – Ifakara Health Institute for their valuable contribution.

## Author contributions

**Conceptualization:** Joseph Barnabas Muganga, Sarah Jane Moore, Mgeni Mohamed Tambwe.

**Data curation:** Masudi Suleiman Maasayi, Mgeni Mohamed Tambwe.

**Formal analysis:** Masudi Suleiman Maasayi, Sarah Jane Moore, Mgeni Mohamed Tambwe.

**Investigation:** Masudi Suleiman Maasayi, Jason Moore, Sarah Jane Moore, Mgeni Mohamed Tambwe.

**Methodology:** Michael Coleman, Neil Francis Lobo, Sarah Jane Moore, Mgeni Mohamed Tambwe.

**Project administration:** Sarah Jane Moore.

**Supervision:** Sarah Jane Moore, Mgeni Mohamed Tambwe.

**Visualization:** Joseph Barnabas Muganga.

**Writing – original draft:** Masudi Suleiman Maasayi.

**Writing – review & editing:** Johnson Kyeba Swai, Jennifer Claire Stevenson, Michael Coleman, Neil Francis Lobo, Sarah Jane Moore, Mgeni Mohamed Tambwe.

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
