## [Decision Letter · Decision Letter 0]

26 Jun 2025

PONE-D-25-08853
Efficacy of the BiteBarrier transfluthrin emanator against susceptible and resistant malaria and arbovirus vectors in the semi-field system in Tanzania
PLOS ONE

Dear Dr. Coleman,

Thank you for submitting your manuscript to PLOS ONE. After careful consideration, we feel that it has merit but does not fully meet PLOS ONE’s publication criteria as it currently stands. Therefore, we invite you to submit a revised version of the manuscript that addresses the points raised during the review process.

Please address the reviewers comments. I think you will agree that most of these suggestions will make the paper stronger. if you wish to rebut any of the rewiewer comments you may do so. 

We look forward to receiving your revised manuscript.

Kind regards,

Stephen M. Rich, MS, PhD

Academic Editor

PLOS ONE

Journal Requirements:

“The authors declare no competing interests.”

4. In the online submission form, you indicated that [Data used to generate conclusions of this study are available from the corresponding author on a reasonable request.].

5. Please amend the manuscript submission data (via Edit Submission) to include author Sarah Jane Moore.

Reviewers' comments:

Reviewer's Responses to Questions

**Comments to the Author**

1. Is the manuscript technically sound, and do the data support the conclusions?

Reviewer #1: Partly

Reviewer #2: Yes

2. Has the statistical analysis been performed appropriately and rigorously? 

Reviewer #1: No

Reviewer #2: Yes

3. Have the authors made all data underlying the findings in their manuscript fully available?

Reviewer #1: No

Reviewer #2: Yes

4. Is the manuscript presented in an intelligible fashion and written in standard English?

Reviewer #1: No

Reviewer #2: Yes

5. Review Comments to the Author

Reviewer #1: The authors performed semi-field system investigations to assay the extent to which the presence of sheets impregnated with transfluthrin might perturb the landing and feeding rates of several kinds of mosquitoes with known susceptibilities to pyrethroids. Spatial repellents and volatile pesticides are increasingly of interest in the public health arena as more traditional residual products are retired from use. Based upon their data, they concluded that the BiteBarrier sheets provide measurable and statistically significant levels of protection in simulated indoor and outdoor conditions.

The study is interesting and could be relevant to the design of certain public health interventions. Whereas the data presented do support conclusions that the treated material reduces landing and feeding rates, the data presentation is incomplete and difficult to fully appreciate, certain potentially confounding factors need to be addressed, and the narrative deserves further attention.

Title: The authors are encouraged to revise the title so that it informs the reader as to what they’ve concluded. The current title simply indicates that they studied the relationship.

Abstract, Line 34: The authors (and this reviewer) understand what is meant by ‘outdoor and early biting mosquitoes’, but many readers would not be so well informed. The authors are encouraged to explain what they mean and describe the significance of the temporal and spatial questing behaviors.

Line 55: The authors might consider rephrasing the first sentence. Whereas it is true that human population movements and environmental changes may enhance transmission of malaria and arboviruses, new tools and strategies are also needed to reduce risk in endemic areas.

Line 120: The authors stated that treatment and control chambers remained fixed throughout the study to avoid cross contamination. To what extent might the transfluthrin have become adsorbed or absorbed by fabrics or other components of the chambers? How certain are the investigators that any such residue would not skew the results of subsequent trials/replicates? Had the investigators assessed landing or feeding rates in treated chambers with the BiteBarrier removed? To what extent might the treated panels in one chamber influence activity within an adjacent one?

Line 121: The number of volunteers (2 for each group) seems low. Did each volunteer rotate between the different study conditions (indoor vs. outdoor, treatment vs. control)? If not, how can the investigators be certain that certain volunteers were not more – or less – attractive to mosquitoes?

Line 137: The text states that four BB sheets were placed in each of the four corners. This was confusing to this reviewer. The Figure seems to indicate that there was just one BB sheet in each corner.

Line 140: How were all mosquitoes collected? Did the collection method cause mortality to any mosquitoes? It would be useful to present the number or proportion of surviving mosquitoes as well as the portion recovered or that had fed.

Line 147: The authors stated that mosquito density was increased to 80 per strain for each replicate. But they stated earlier (Line 141) that 80 mosquitoes of each strain were used. This is confusing.

Line 154: Did the investigators measure wind speed and direction? If so, how? Where were the volunteers situated relative to the BBs and prevailing air flow?

Line 156: Why were only 40 mosquitoes released for the outdoor tests?

Line 206: What amount of blood within a mosquito would qualify it to be considered as fed? How was the extent of bloodfeeding assessed?

Line 215: The description of protection efficacy is confusing to this reviewer, and likely would be beyond comprehension by readers not familiar with landing and feeding assays.

Table 1: The actual numbers of mosquitoes landing or feeding should ideally be displayed.

Table 2: The authors have seemingly provided an aggregate of the number of recovered and bloodfed mosquitoes across all replicates. How much variation occurred between replicates? Were some volunteers more attractive than others to mosquitoes?

Line 254: This sentence seems to imply that there was greater mortality among pyrethroid-resistant mosquitoes. Table 3 suggests otherwise.

Line 285: Mosquitoes can transmit pathogens, but they cannot transmit diseases. The authors are encouraged to avoid colloquial phrasing regarding ‘disease transmission’ by mosquitoes. Infectious mosquitoes that are probing and salivating may transmit arboviruses as well as sporozoites of Plasmodium. Accordingly, a landing rate may be of greatest significance to assessments of transmission of pathogens from mosquitoes to people. In contrast, feeding rates are more relevant to assessments of uptake of pathogens by mosquitoes. The authors are encouraged to elaborate on the discussion of their data as they relate to these features of transmission dynamics.

Line 292: The authors state that recovery rates were ‘below or exceeded the estimated release mosquito count’. This statement is troubling to this reviewer, as the authors mentioned earlier that they released 40 or 80 mosquitoes per replicate. It should have been easy and straightforward to precisely count the 40 or 80 mosquitoes for release. It is understandable that a few released mosquitoes might have been elusive and not recovered at the end of each replicate. When the recovery rates exceeded the estimated release count, could the authors comment on whether the recovered mosquitoes among living or dead? Did they use different colored fluorescent tags that might allow them to assess the species or group to which they belonged? Perhaps, the authors might have increased the interval between the replicates to better ensure that live mosquitoes released during an earlier trial would not confound results of a subsequent replicate.

Line 336: The authors acknowledge the need for further investigations beyond eight weeks of product aging. Why was the eight-week interval selected for this study? What is the expected duration useful life of this product?

Figures: These do not impart sufficient information to justify their inclusion.

Reviewer #2: This study evaluates the protective efficacy of the BiteBarrier transfluthrin emanator against malaria-transmitting vectors. The choice of mosquito species and the study duration represent important contributions to the field. The use of blood-feeding reduction as an indicator of protection—and its comparison to landing-based assessments—raises relevant questions about well established methods such as HLC.

To enhance the reader’s understanding of the environmental context, we recommend including photographs of the study sites. The current Figure 1 and its accompanying description do not adequately depict key aspects of the setting, such as surrounding vegetation and wind barriers. These features are critical, as wind conditions—discussed in the manuscript—can substantially influence both the performance of spatial repellents and mosquito behavior.

Additionally, we suggest that the authors expand the definition of protective efficacy, specifying how it was calculated and applied within the study. In particular, it would be helpful to clarify how the relative protective efficacy was derived from the comparison of mosquito landing and feeding outcomes. This clarification is essential for a full understanding of the results presented in Table 1.

6. PLOS authors have the option to publish the peer review history of their article (what does this mean?). If published, this will include your full peer review and any attached files.

Reviewer #1: **Yes: **Richard Pollack

Reviewer #2: **Yes: **Sebastián D'hers

---

## [Author Response · Author response to Decision Letter 1]

18 Aug 2025

See response to reviewers letter, no additional notes.

---

## [Editor Report · Decision Letter 1]

8 Sep 2025

The BiteBarrier transfluthrin emanator demonstrates significant protection against susceptible and resistant malaria and arbovirus vectors in semi-field trials in Tanzania

PONE-D-25-08853R1

Dear Dr. Coleman,

We’re pleased to inform you that your manuscript has been judged scientifically suitable for publication and will be formally accepted for publication once it meets all outstanding technical requirements.

Kind regards,

Stephen M. Rich, MS, PhD

Academic Editor

PLOS ONE

---

## [Editor Report · Acceptance letter]

PONE-D-25-08853R1

PLOS ONE

Dear Dr. Coleman,

I'm pleased to inform you that your manuscript has been deemed suitable for publication in PLOS ONE. Congratulations! Your manuscript is now being handed over to our production team.

Kind regards,

on behalf of

Dr. Stephen M. Rich

Academic Editor

PLOS ONE